# Cold Posterior Effect towards Adversarial Robustness

**Bruce Rushing**
Department of Philosophy
Purdue University
West Lafayette, IN 47907
bmrushin@purdue.edu

**Antonios Alexos**
Department of Computer Science
University of California, Irvine
Irvine, CA 92697
aalexos@uci.edu

**Harrison Espino**
Department of Computer Science
University of California, Irvine
Irvine, CA 92697
espinoh@uci.edu

**Nicholas Cohen**
Department of Computer Science
University of California, Irvine
Irvine, CA 92697
cohenn1@uci.edu

**Pierre Baldi**
Department of Computer Science
University of California, Irvine
Irvine, CA 92697
pfbaldi@uci.edu

## Abstract

In this study, we delve into the application of Bayesian Neural Networks (BNN) as a prominent strategy for addressing adversarial attacks, elucidating their enhanced robustness. Specifically, our investigation centers on the cold posterior effect within BNNs and its role in fortifying the models against adversarial perturbations. Our findings underscore that harnessing the cold posterior effect markedly augments the models' resilience to adversarial manipulations when compared to warm counterparts, thereby elevating the overall security and reliability of the model. To substantiate these observations, we conduct rigorous experiments involving popular white-box and black-box attacks, on both fully connected networks and ResNet-20 architectures. Our empirical results unequivocally demonstrate the superior performance of cold models over warm models with multiple training methods including SGMCMC, SGHMC, and VI, against adversarial threats in diverse scenarios. This study not only contributes empirical evidence but also offers theoretical insights elucidating the efficacy of the cold posterior effect in bolstering the adversarial robustness of BNNs.

## 1 Introduction

Bayesian Neural Networks (BNNs) are a type of Bayesian Model Averaging over neural network parameters that gives them improved generalization performance thanks to a natural regularization effect on their predictions [Wilson and Izmailov, 2020, Jospin et al., 2022, Alexos et al., 2022]. This makes them useful for tasks such as hyperparameter optimization [Wu et al., 2019, Feurer et al., 2015]. One desirable feature of BNNs is their robustness to adversarial attacks [Gal and Smith, 2018]. White-box attacks, such as Fast Gradient Sign Method (FGSM) and Projected Gradient Descent (PGD), that exploit model gradients do not diminish BNNs' accuracy in classification tasks as sharply as non-Bayesian networks [Uchendu et al., 2021]. One likely reason for this robustness

Workshop on Bayesian Decision-making and Uncertainty, 38th Conference on Neural Information Processing Systems (NeurIPS 2024).

is due to BNNs conduct inference via weighted model averaging by a posterior, which confers desirable gradient properties in the over-parameterized limit [Carbone et al., 2020]. BNN experiments often show that models that use a modified, tempered posterior perform better in classification tasks [Wenzel et al., 2020]. Such tempered posteriors—also called *cold posteriors*—have the effect of artificially sharpening the model average around its mode, and the empirical results have found tempering close to zero performs better regarding accuracy; hence naming this *the cold posterior effect*. Considerable work has been done to find the reason for this cold posterior effect [Noci et al., 2021, Fortuin et al., 2021]. Although much work has been done on BNNs and adversarial robustness [Carbone et al., 2020, Panousis et al., 2021, Wicker et al., 2021], there has been little work looking at the cold posterior effect and BNN adversarial robustness together. Several empirical and theoretical questions remain open: Are the BNNs that are more robust to adversarial attacks also models with cold posteriors? Does the cold posterior effect increase or decrease when doing adversarial attacks? If it does, why does it change? Is there a connection between the cold posterior effect and BNN adversarial robustness?

## 2    Analysis of Cold Posteriors on Adversarial Attacks

Following Carbone et al. [2020] and supposing an uninformative prior over model parameters, we can show that the expected gradient on that prior is zero:

**Proposition 2.1.** *Let $f(\mathbf{x}, \theta)$ be a fully trained overparameterized BNN on a prediction problem with data manifold $\mathcal{M}_D \subset \mathbb{R}^d$ and posterior weight distribution $\frac{p(\theta|D)^{1/T}}{Z(p)}$ with a uniform prior and renormalization constant $Z(p)$ and let $T \to 0$. Assuming $\mathcal{M}_D \in \mathcal{C}^\infty$ almost everywhere, in the large data limit we have a.e. on $\mathcal{M}_D$:*

$$(\langle \nabla_{\mathbf{x}} L(\mathbf{x}, \theta) \rangle_{\frac{p(\theta|D)^{1/T}}{Z(p)}}) = \mathbf{0}$$

*Proof.* The proof proceeds the same as in Carbone et al. [2020]. Note that the only difference with the tempered posterior is that:

$$p(\theta|D)^{1/T} = \frac{p(D|\theta)^{1/T} p(\theta)^{1/T}}{(\sum_{\theta^*} p(D|\theta^*) p(\theta^*))^{1/T}}$$

Since for any $\theta, \theta'$, $p(\theta) = p(\theta')$, it follows that $p(\theta)^{1/T} = p(\theta')^{1/T}$ and so the tempered posterior for will be identical for all weights $\theta$. It follows from Carbone et al. [2020] Lemma 2 that gradients will cancel out, resulting in the expected gradient to be $\mathbf{0}$.  □

This shows that tempering should lead to the same robustness against adversarial attacks as the "untempered", or warm, posterior. In the overparameterized limit, the expected gradient under a uniform prior will cancel, and there will be no direction to exploit by white-box attacks on the Bayesian model.

## 3    Experiments and Discussion

To test the cold posterior effect and its extent under adversarials, we trained BNNs with stochastic gradient Markov Chain Monte Carlo (SGMCMC) with stochastic gradient Langevin Dynamics (SGLD) and Hamiltonian Monte Carlo (SGHMC) and Variational Inference (VI) on MNIST and CIFAR-10 datasets. Our primary experiments were on fully connected network (FCN) BNNs with two hidden layers of 128 and 64 nodes respectively and a Resnet-20. Our prior probabilities for the hidden dense BNNs and the Resnet-20 were the standard normal, $\mathcal{N}(0, \mathbb{I})$. For SGMCMC, we used cyclical restarts as practiced in Zhang et al. [2019], Wenzel et al. [2020] and we collected 12 samples across 28 epochs and 4 restarts on both SGLD and SGHMC with a burn-in of 3 epochs; for VI, we employed Bayes by Backprop (BBB) with the Adam optimizer and stepped learning rates. With SGHMC, we found a friction parameter of $\eta = 0.1$ to perform best used it in experiments. We tested the cold posterior effect on temperatures between $1.0$ and $0.00001$. Since data augmentation is correlated with the cold posterior effect [Izmailov et al., 2021], we did not train the FCN on MNIST in

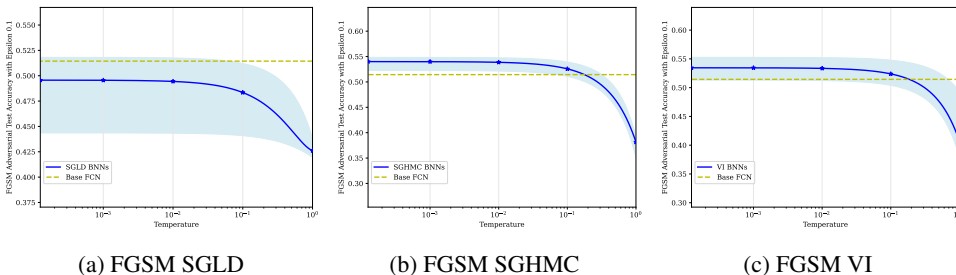

(a) FGSM SGLD                    (b) FGSM SGHMC                    (c) FGSM VI

Figure 1: The cold posterior effect in FGSM attacks on MNIST with $\epsilon = 0.1$ and 12 samples drawn from the posterior $p(\theta|\mathcal{D})$. The blue line indicates the best degree 2 polynomial fit to observed accuracies, and the light blue shaded region indicates the range of observed accuracy values. Temperatures $T \ll 1$ were found to perform best.

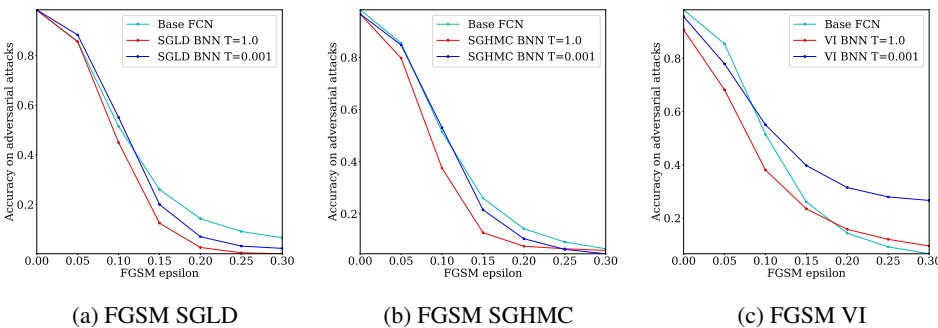

(a) FGSM SGLD                    (b) FGSM SGHMC                    (c) FGSM VI

Figure 2: The effects of FGSM adversarial attacks on accuracy for warm and cold FCN BNNs on MNIST. VI shows that cold models degrade more gracefully and the extent of the cold posterior effect grows with stronger attacks, while SGLD and SGHMC perform better at weaker attacks, where they show a greater cold posterior effect.

both SGMCMC and VI with data augmentation. For boosting the accuracy of Resnet20 on CIFAR-10, we used data augmentation proposed by Cubuk et al. [2019].

For the white-box adversarial attacks, we leveraged the FGSM and PGD attacks. The FGSM and PGD attacks were computed across a holdout test set for MNIST with the holdout test set of $\{1000, 10000\}$ samples, but for CIFAR-10 we used the whole test set of 10000 samples.

## 3.1   Fully Connected Network on MNIST

We observe a pronounced cold posterior effect on fixed $\epsilon = 0.1$ across temperatures $T \in \{0.0001, 1.0\}$. fig. 1 show this cold posterior effect for FGSM (see the appendix for PGD). Across SGLD, SGHMC, and VI we found $T = 1.0$ models to underperform the non-Bayesian model; however, models with $T \ll 1$ consistently beat the FCN and displayed a dramatic cold posterior effect. SGLD performed worse than the traditional FCN, while SGHMC and VI showed adversarial robustness at colder temperatures. In PGD, we see adversarial robustness from cold models across SGLD and SGHMC relative to the baseline FCN.

To test whether the cold posterior effect grows with adversarial perturbation, we examined two models with $T = 1.0$ and $T = 0.001$ and conducted adversarial attacks across $\epsilon \in \{0.0, 0.05, 0.1, 0.15, 0.2, 0.25, 0.3\}$ for FGSM and PGD. fig. 2 shows the increase in the cold posterior effect as the $\epsilon$ increases. The effect depends on the type of BNN; with VI, the effect is most pronounced as $\epsilon$ increases, while both SGLD and SGHMC see the biggest gaps at lower $\epsilon$s. VI shows that colder models degrade more gracefully across FGSM attacks. We find the baseline model performs similarly to the BNNs up to $\epsilon = 0.1$, after which it suffers significantly from the adversarial attack. Again, PGD attacks of 5 and 20 iterations with $\alpha = 0.1$ (see appendix) showed that cold posteriors substantially improve robustness. At 5 iterations of PGD, the performance of all models

**Cold Posterior Effect of FGSM and PGD attacks on Resnet-20**

| Method and Attack | Temperature | | | | |
|---|---|---|---|---|---|
| | **0.0001** | **0.001** | **0.01** | **0.1** | **1.0** |
| SGLD-FGSM | **23.89** | 22.9 | 20.67 | 17.19 | 11.12 |
| SGLD-PGD | 0.31 | **0.86** | 0.25 | 0.09 | 0 |
| SGHMC-FGSM | 20.28 | 19.65 | **21.42** | 18.72 | 9.77 |
| SGHMC-PGD | 0.38 | **0.39** | 0.4 | 0.11 | 0 |
| VI-PGD | **20.17** | 4.2 | 4.46 | 2.89 | 2.38 |
| VI-FGSM | 31.9 | **32.94** | 32.06 | 31.54 | 25.71 |

Table 1: The effects of PGD and FGSM adversarial attacks on accuracy for warm and cold Bayesian Resnet-20 models on CIFAR-10. Highest accuracy in each row is highlighted in bold. We observe that the cold models outperform the warm models.

steadily declines as epsilon is increased. Notably, the accuracy of the warm model declines faster than the cold model, resulting in a larger cold posterior effect. This is much more pronounced at 20 iterations, in which all models' performance declines faster and the accuracy difference between the warm and cold models widens quicker.

## 3.2   Resnet-20 on CIFAR-10

To test the cold posterior effect for adversarial robustness in real-world data, we conducted adversarial attack experiments on CIFAR-10 with Resnet-20. More specifically, we examined Resnet-20 trained via VI, SGLD, and SGHMC with $T \in \{1.0, 0.1, 0.01, 0.001, 0.0001\}$ and conducted adversarial attacks across $\epsilon \in \{0.0, 0.05, 0.1, 0.15, 0.2, 0.25, 0.3\}$ for white-box attacks PGD and FGSM. For VI we used 64 posterior samples for the inference of the model, and for SGLD and SGHMC we used 6 samples. All the images from the official test set of CIFAR10 which are 10,000. The main results can be seen in table 1 where we show the results with $\epsilon = 0.1$. Although the models with all temperatures suffer significantly from adversarial attacks, we observe that the colder models perform better than the warm model, and the colder the model the better its performance towards adversarial attacks. This effect is more profound on the FGSM attack, and although the PGD attack experiments showed too much accuracy degradation there is still some cold posterior effect. We speculate the bad performance is because we did not use standard adversarial defense techniques or mechanisms for our experiments.

We also conducted black-box attacks, and we focused on the Salt and Pepper attack. Our results show improved adversarial robustness with cold models in the appendix. All BNN types showed this effect, but notably unlike our earlier results on white-box attacks, VI showed the weakest performance. SGLD showed the strongest with SGHMC very close. SGLD showed a large performance gain in its accuracy over the warm model. ResNet-20 performed better against the Salt and Pepper attack compared to PGD and FGSM attacks since black-box attacks are easier to defend against because the attack does not have access to the model Chakraborty et al. [2021].

## 3.3   Adversarial Training

We extended the examination of our cold posterior theory to assess its efficacy in enhancing adversarial robustness through additional adversarial defense strategies, specifically employing adversarial training on Resnet-20. Resnet-20 underwent training using PGD adversarial training [Madry et al., 2017] on the CIFAR-10 dataset. Subsequently, the model was subjected to testing under PGD attacks as previously. For VI, we conducted 7 iterations of PGD training, while for SGMCMC and SGHMC, only 2 iterations were performed due to suboptimal training with more iterations. The noise levels for the PGD attacks were $\epsilon \in \{0.05, 0.1, 0.15, 0.2, 0.25, 0.3\}$. Our results from this experiment demonstrate consistent outperformance of cold posterior models over the warm posterior counterpart for all training methods. Notably, the VI-trained model with a temperature

parameter $T = 0.1$ exhibited superior performance compared to other cold posterior models, contrary to observations in non-adversarial experiments from the literature, where colder models generally lead to improved performance. In SGMCMC and SGHMC we saw that colder models performed better. This discrepancy warrants further investigation into the nuanced interplay between adversarial training and temperature parameters within the context of the cold posterior theory.

## 4 Conclusions

In this work, we explored the cold posterior effect in white-box and black-box adversarial attack setups where the cold posterior models outperform the warm posterior models against these attacks. This was shown across multiple BNN model types including SGLD, SGHMC, and VI by BBB in both fully connected and Resnet architectures on MNIST and CIFAR-10 datasets. We show that the cold posterior effect grows as we increase the intensity of the adversarial attacks. We have found that the cold posterior effect is still present with adversarial training, and that the gradients of the cold posterior model vanish slower than of the gradients of the warm posterior model.

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

## Appendix A

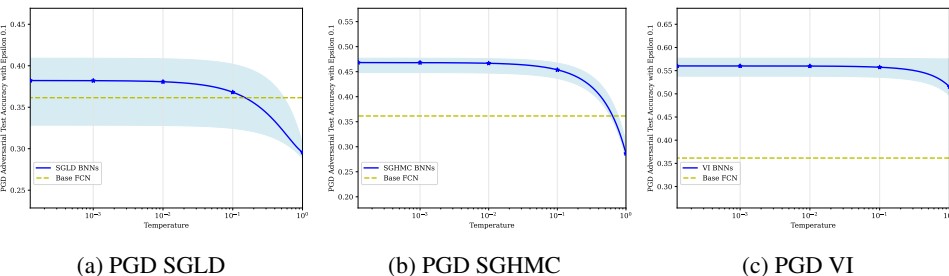

(a) PGD SGLD  (b) PGD SGHMC  (c) PGD VI

Figure 3: The cold posterior effect in PGD attacks on MNIST with $\epsilon = 0.1$ and 16 samples drawn from the posterior $p(\theta|\mathcal{D})$. PGD was ran for 10 iterations. The blue line indicates the best degree 2 polynomial fit to observed accuracies, and the light blue shaded region indicates the range of observed accuracy values concerning temperature. $T \ll 1$ were found to perform best.

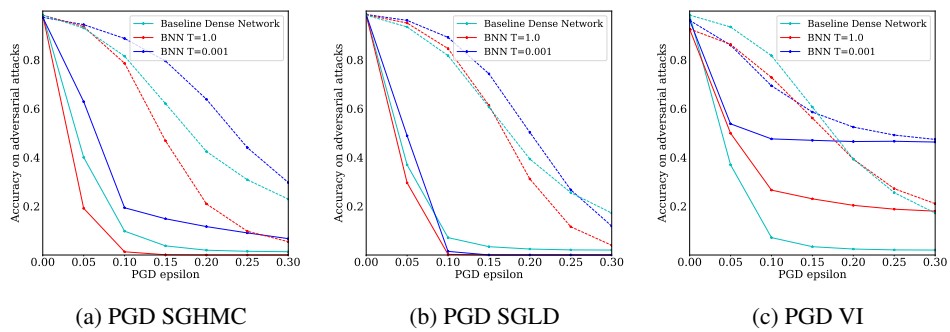

(a) PGD SGHMC  (b) PGD SGLD  (c) PGD VI

Figure 4: The effects of PGD adversarial attacks on accuracy for warm and cold fully-connected BNNs on MNIST. Solid lines indicate 20 iterations of PGD, dashed lines indicate 5 iterations. Warm model accuracy declines faster with increasing epsilon, increasing the extent of the cold posterior effect.

**Cold Posterior Effect of Salt and Pepper Attacks on Resnet-20**

| Method and Attack | Temperature | | | | |
|---|---|---|---|---|---|
| | **0.0001** | **0.001** | **0.01** | **0.1** | **1.0** |
| VI-PepperSalt | 14.8 | 14 | 10.4 | **33.2** | 11.2 |
| SGLD-PepperSalt | 94.4 | 94 | 94.4 | **94.8** | 84.8 |
| SGHMC-PepperSalt | **94** | 93 | **94** | **94** | 91 |

Table 2: The effects of Salt and Pepper adversarial attack on accuracy for warm and cold Bayesian Resnet-20 models on CIFAR-10. Highest accuracy is highlighted in bold. We observe that the cold models outperform the warm models.

