# OpenReview forum: "Cold Posterior Effect towards Adversarial Robustness"
_NeurIPS.cc/2024/Workshop/BDU — NeurIPS BDU Workshop 2024 Poster_

### Official Review · Reviewer_P765 · 2024-09-14

**Rating:** 5
**Confidence:** 4

**Review:**

**Summary:**
This paper explores the cold posterior effect in BNNs and its role in enhancing adversarial robustness. Through extensive experiments, the results demonstrate that cold posterior models consistently outperform warm counterparts in resisting adversarial attacks.

**Strengths:**
1. The paper explores the "cold posterior effect" in enhancing adversarial robustness, combining this effect with Bayesian Neural Networks to improve adversarial defense.
2. The paper conducts a comprehensive experiments on MNIST and CIFAR-10 datasets using various models effectively validating the impact of different posterior temperatures on adversarial robustness.

**Opportunities For Improvement:**
1. Some claims in the paper are inconsistent. In the adversarial training section, the authors state that the VI method at T=0.1 outperforms other cold posterior models, but figures 2 and 4 show that this method outperforms not only in adversarial training but also in other scenarios compared to the other models.
2. I want to know why the VI method generally performs better than other approaches. Why is the cold posterior method more effective? Moreover, the paper does not analyze how common adversarial defense techniques can be combined with the cold posterior approach.
3. Although several experimental results are presented, the paper lacks significance testing or statistical analysis to confirm the robustness of these findings. For example, it is unclear whether the performance differences between models at different temperatures are statistically significant.

**Minor Typos:**
In Table 1, there should be no text in the headers, and for consistency, the last two rows should be swapped, placing FGSM above PGD.

---

### Official Review · Reviewer_gkWZ · 2024-10-01
**The paper effectively demonstrates the advantages of cold posteriors in Bayesian Neural Networks for enhancing adversarial robustness, though it could benefit from more comprehensive theoretical analysis and broader experimental validation.**

**Rating:** 8
**Confidence:** 4

**Review:**

This paper presents a compelling exploration of the cold posterior effect in Bayesian Neural Networks (BNNs) and its impact on adversarial robustness. The authors conduct thorough experiments using various training methods, including SGMCMC, SGHMC, and VI, on both fully connected networks and ResNet-20 architectures. The empirical results clearly demonstrate that cold models outperform warm models in terms of resilience to adversarial attacks, across different scenarios and datasets. The study not only provides valuable empirical evidence but also offers theoretical insights into why the cold posterior effect enhances the adversarial robustness of BNNs.

However, the paper has some limitations. Firstly, the theoretical explanations for the cold posterior effect could be more detailed and rigorous. While the authors provide some propositions and proofs, a deeper theoretical analysis would strengthen the argument. Secondly, the experimental setup could be improved by including more datasets and attack types to validate the generalizability of the findings. Additionally, the choice of parameters for different training methods lacks detailed explanation, which might make the results less reproducible.

---

### Decision · Program_Chairs · 2024-10-09

Accept (Poster)